# DNA damage and antioxidant capacity in COPD patients with and without lung cancer

**Camila Freitas dos Santos**[1¤]*, **Mariana Gobbo Braz**[2], **Nayara Micarelli de Arruda**[2‡], **Laura Caram**[3‡], **Duelene Ludimila Nogueira**[3‡], **Suzana Erico Tanni**[3], **Irma de Godoy**[3], **Renata Ferrari**[3]

**1** UNINOVE, Bauru Campus, Bauru, Brazil, **2** Botucatu Medical School, GENOTOX Laboratory, São Paulo State University—UNESP, São Paulo, Brazil, **3** Department of Internal Medicine, Botucatu Medical School, São Paulo State University—UNESP, São Paulo, Brazil

☯ These authors contributed equally to this work.
¤ Current address: Department of Health 3, UNINOVE, Bauru, São Paulo, Brazil
‡ NMA, LC and DLN also contributed equally to this work.
* fsantos.cami@gmail.com

## Abstract

### Background and objective

Chronic obstructive pulmonary disease (COPD) is characterized by chronic inflammation of the lower airways, and COPD patients show two to five times higher risk of lung cancer than smokers with normal lung function. COPD is associated with increased oxidative stress, which may cause DNA damage and lung carcinogenesis. Our aim was to evaluate DNA damage and oxidative stress (lipid peroxidation and antioxidant status) and their relationship in patients with COPD with and without lung cancer.

### Methods

We evaluated 18 patients with COPD, 18 with COPD with lung cancer, and 18 controls (former or current smokers). DNA damage was evaluated in peripheral blood lymphocytes using a comet assay; the concentration of malondialdehyde (MDA) and hydrophilic antioxidant performance (HAP) were measured in the plasma.

### Results

DNA damage was higher in patients with COPD with cancer than in the controls (p = 0.003). HAP was significantly lower in patients with COPD with cancer than in those without cancer and controls. The presence of lung cancer and COPD showed a positive association with DNA strand breaks and the concentration of MDA.

### Conclusion

COPD with lung cancer was associated with elevated DNA damage in peripheral lymphocytes, and cancer and COPD showed a positive correlation with DNA damage. The antioxidant capacity showed a negative association with the interaction COPD and cancer and presence of COPD. The mechanisms underlying the increased incidence of lung cancer in

**Data Availability Statement:** The data cannot be shared due to potentially identifying or sensitive information. Future researchers will be able to request the data used in this study through the

contact of the local ethics committee of São Paulo State University: Chácara Butignolli, s / n, Botucatu, São Paulo - Brazil. Zip code: 18618970. telephone: 3880-1609. email: cep.fmb@unesp.br. Additionally, the study ethics approval may be requested through the conep portal (https://Plataformabrasil.saude.gov.br/).

**Funding:** Renata Ferrari: This work was supported by the National Council for Scientific and Technological Development (Conselho Nacional de Desenvolvimento Científico e Tecnológico - CNPq) [grant numbers 470496/2014-2]. Camila Freitas dos Santos: This work was supported by São Paulo Research Foundation (Fundação de Amparo à Pesquisa do Estado de São Paulo - FAPESP) [grant numbers 2019/15075-2].

**Competing interests:** The authors have declared that no competing interests exist.

COPD are unknown; DNA damage may be involved. Further research may provide insights into their development and treatment.

## Introduction

Chronic obstructive pulmonary disease (COPD) is a leading cause of morbidity and mortality worldwide as reported by the World Health Organization [1]. The disease is predicted to become the third cause of death worldwide by 2060 with approximately 5.4 million annual deaths [2]. In addition, patients with COPD have two to five times higher risk for developing lung cancer than smokers without obstructive disease [3]. A systematic review analysed 21 articles from different electronic databases to determine the prevalence of lung cancer in patients with COPD. Lung cancer prevalence was 15.3% in a population of 11898 participants (2,309 with COPD and 9,589 controls with normal spirometry); patients with COPD were 6.35 times more likely to have lung cancer than controls [4].

Smoking is the common risk factor for the development of diseases, such as COPD and lung cancer [5]. Cigarettes have free radicals in their composition, including reactive nitrogen and oxygen species (RNOS). Oxidative stress leads to the degradation of tumour suppressor proteins, increased cell division, and decreased DNA repair [6]. Furthermore, DNA damage may lead to genetic mutation, which when combined with continued exposure to the RNOS, may contribute to the development of cancer. Smokers and patients with COPD present shortened peripheral lymphocyte telomeres, associated with a decreased cell life span, when compared to healthy individuals, which makes them more susceptible to cancer [7, 8]. Chronic inflammation is one of the common mechanisms associated with both COPD and lung cancer since it is associated with a malignant transformation from repeated airway epithelial injury and cell turnover with the propagation of DNA mutations caused by cigarette exposure [3].

Oxidative stress markers are identified as possible diagnostic predictors of cancer [9]. It has already been reported higher malondialdehyde (MDA) level, an important lipid peroxidation marker, in lung cancer patients than in controls [9]. Decreased antioxidant status suggests a link between oxidative stress and malignant transformation; it has been shown a relationship between MDA levels and more advanced clinical stages of the disease, which could be used for staging [10].

DNA damage in patients with COPD with and without lung cancer is not yet clearly established in the literature. Although there are reports regarding this relationship, [3–7, 11–15] its influence is unknown in patients with lung cancer when compared to patients with pre-existing COPD.

We hypothesized that patients with COPD and lung cancer have increased DNA damage and oxidative stress when compared to patients without cancer and smokers. Thus, the purpose of this study was to assess DNA damage and oxidative stress in patients with COPD with and without lung cancer and former or current smokers (controls).

## Materials and methods

### Study population

Eighty individuals from the Pulmonology Outpatient Unit at Botucatu Medical School were recruited, and 54 were included in this cross-sectional study (Fig 1). The inclusion criteria included a post-bronchodilator $FEV_1/FVC$ ratio (ratio of the forced expiratory volume in the first second to the forced vital capacity) of less than 0.7 [2]. The exclusion criteria included the

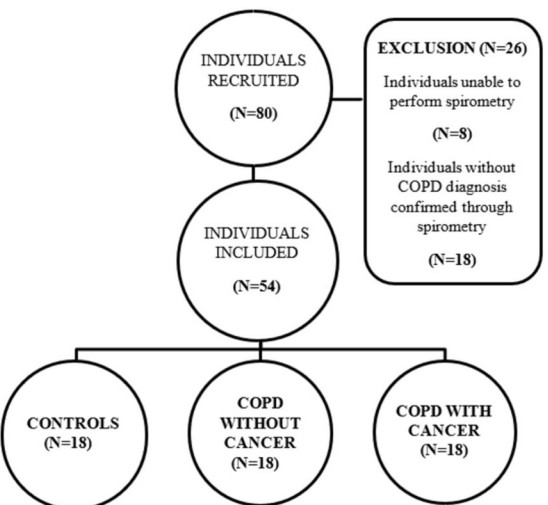

**Fig 1. Sample selection and reasons for patient exclusion.**

presence of neoplasms other than lung cancer, history of lung resection, history of drug abuse, history of exposure to radiation, history of use of antioxidants or vitamins, presence of lung tuberculosis or any other active infection, and presence of acute or chronic pulmonary disease other than COPD. Individuals who were active smokers, without comorbidities and with spirometry tests results within the normal range were included in the control group.

Participants were made aware of the proposed study procedures and written informed consent was obtained. All of the procedures were approved by the Research Ethics Committee of Botucatu Medical School University Hospital, Brazil (CAAE N° 49692515.1.0000.5411).

## Pulmonary function, pulse oximetry, smoking status, and dyspnoea score

Pre- and post-bronchodilator spirometry were performed using a KOKO spirometer (Ferrari KOKO Louisville, CO 80027, USA) according to the criteria set by the American Thoracic Society [16]. $FEV_1$ values were expressed in litres and as percentages of the FVC and reference values [17]. Pulse oximetry was conducted using an Onyx oximeter (Model 9500 Oximeter, Nonin Medical Inc., Minneapolis, MN, USA) while the patients were breathing room air. Smoking history and current smoking status were investigated and a confirmation of smoking status was performed by measuring carbon monoxide (CO) in the exhaled air using a standardized technique (Micro CO Meter, ©Cardinal Health, England, UK). An exhaled CO value > 6.0 ppm was considered active smoking [18, 19]. Dyspnoea was assessed using a translated version of the Modified Medical Research Council scale [20].

## Venous blood collection and evaluation of DNA damage and oxidative stress markers

Fasting peripheral blood samples were collected in the early morning (between 8 and 10 am). The plasma and isolated lymphocytes (conducted under red light) were stored at -80°C until the analyses. From the blood collection until assays were performed, all steps were performed under yellow light to avoid possible bias for oxidative stress and DNA damage. Coded samples were analysed for all biomarkers in a blind manner.

The comet assay was conducted in duplicate according to Singh et al. [21] and Tice et al. [22] with slight modifications [23]. Briefly, 10-μL lymphocytes were embedded into low

melting point agarose and spread on agarose-precoated microscope slides. The slides were immersed overnight in a freshly prepared cold lysing solution (2.5 M NaCl, 100 mM ethylene-diaminetetraacetic acid [EDTA], 10 mM Tris, pH 10.0, with a freshly prepared solution of 1% Triton X-100 and 10% dimethylsulfoxide) at 4˚C. After 5 min washing in phosphate-buffered saline, the slides were immersed in an alkaline buffer (0.3 M NaOH and 1 mM EDTA at pH > 13) in a horizontal tank for 20 min to allow DNA unwinding before the 20-min electrophoresis period conducted at 25V and 300 mA (in the same solution at 4˚C). After electrophoresis, the slides were neutralized (0.4 M Tris, pH 7.5), stained with 50 μL SYBR Gold, and analysed in a fluorescence microscope at 400× magnification, using an image analysis system (Comet Assay IV—Perceptive Instruments, Suffolk, UK). One hundred randomly selected nucleoids/participants (50 from each of two replicated slides) were evaluated, and the tail intensity (% DNA Tail) was considered.

To evaluate oxidative stress, both MDA, a gold-standard marker of lipid peroxidation [24], and hydrophilic antioxidant performance (HAP) that measures the antioxidant capacity in the aqueous compartment of plasma to assess antioxidant defense, which clears free radical/RNOS activity and accumulation in cells and maintain redox balance [25], were evaluated as biomarkers.

A 100-μL aliquot of the plasma was used for the MDA concentration analysis in duplicate, according to a previous protocol [26], with slight modifications. Briefly, we add 700 μL of 1% orthophosphoric acid and 200 μL of thiobarbituric acid (42 mM) to the sample and then heated it for 60 min in a water bath at 100˚C; the sample was cooled on ice immediately after. Two hundred microliters were transferred to a 2-mL tube containing 200 μL of NaOH-methanol (1:12 v/v). The sample was vortex-mixed for 10 s and centrifuged for 3 min at 13000 g. We transferred 200 μL of the supernatant to a 300-μL glass vial and injected 50 μL onto the column. Shimadzu LC-10D system (Shimadzu Corporation, Kyoto, Japan) was used for high-performance liquid chromatography, with a C18 Luna column (5 μm, 150 × 4.60 mm, Phenomenex Inc., Torrance, CA, USA), a Shimadzu RF-535 fluorescence detector (excitation: 525 nm, emission 551 nm), and 0.5 mL/min flow of $KH_2PO_4$ (1 mM, pH 6.8). MDA was quantified by the area determination of the peaks in the chromatograms relative to a standard curve of known concentrations [27].

For the antioxidant analysis, the HAP (%) was determined fluorometrically, in triplicate, as described by Beretta et al. [28] with slight modifications, [29] using the VICTOR X2 2030 Multilabel Reader (PerkinElmer, Boston, USA). The plasma antioxidant capacity was quantified by comparing the area under the curve relating to the oxidation kinetics of the suspension phosphatidylcholine (PC), which was used as a reference biological matrix. The 2,2′-azobis-(amidinopropane) dihydrochloride was used as a peroxyl radical initiator. The results represented the percent inhibition (4,4-difluoro-5-[4-phenyl-1,3-butadienyl]-4-bora-3a,4a-diaza-s-indacene-3-pentanoic acid)—(BODIPY) in 581/591 plasma samples with respect to the findings in the control BODIPY 581/591 PC liposome samples.

## Statistical analysis

Sample size was calculated for a multiple linear regression effect size of 0.3 with an estimated multiple correlation squared of 0.50 with the addition of four predictors in the model (G Power 3.1.3).

Descriptive statistics were used to describe the features of all participants. Means ± standard deviations or medians and interquartile ranges (25–75%) were used depending on data distribution. Categorical variables are expressed as percentages. The chi-square test was used to compare the values of categorical variables.

The analysis of variance followed by the Tukey or Kruskal–Wallis test followed by Dunn's test was used to compare patient characteristics.

The multiple linear regression analysis was used to assess associations between the interaction variables (COPDxlung cancer), COPD (absence = 0 and presence = 1), lung cancer (absence = 0 and presence = 1), the antioxidant capacity, oxidative stress markers, and DNA damage in the three study groups. Setting variables in the model were sex and age. Collinearity was prevented by deleting a variable showing correlation. The level of significance was set at 5%. All analyses were performed using IBM SPSS Statistics 23 and Sigma Plot 11.0.

## Results

Table 1 shows the general characteristics of the 54 participants enrolled in the study and the comparison between groups. The controls were younger and had higher body mass index (BMI) than patients with COPD with and without lung cancer. Patients with lung cancer had a higher incidence of smoking history than the controls.

Fig 2 shows the level of DNA damage in the analysed groups. Patients with COPD and lung cancer presented higher values than the controls ($p < 0.05$). There were no statistically significant differences between the controls and the patients with COPD without cancer, and between patients with COPD with and without lung cancer.

MDA levels were not statistically significant different between the controls (4.9 μmol/L [4.4–5.8]), patients with COPD without cancer (4.5 [3.7–6.7]), and patients with COPD with lung cancer (5.5 μmol/L [4.3–8.9]).

Fig 3 shows the concentration of the antioxidant capacity in the three groups. Patients with COPD with lung cancer presented lower values than controls and patients with COPD without lung cancer ($p < 0.05$). There was no statistically significant difference between controls and patients with COPD without lung cancer.

The multiple linear regression analysis identified the interaction between COPD and lung cancer as independent predictors of MDA levels, HAP, and DNA strand breaks (Table 2). HAP also showed a statistically negative association with COPD presence, but the other variables did not show any association with it.

## Discussion

Our findings showed that patients with COPD and lung cancer presented increased levels of DNA damage determined by the comet assay when compared to the controls. The percentage of HAP was lower in patients who had an association of lung cancer and COPD, than in those in the other two groups. DNA strand breaks and MDA levels showed a positive association with interaction COPD and cancer in the multiple linear regression analysis; in contrast, HAP showed a negative association.

In our sample, there was no significant difference in smoking history between the COPD and COPD cancer groups, but there was a difference between the control and COPD cancer groups. Based on this information, it is possible to relate the higher levels of DNA damage in the COPD group with the greater exposure to cigarettes, given the greater number of pack-years, as shown in Table 1.

The relationship established between COPD and lung cancer is based on similar mechanisms of local immune response owing to cigarette use in both diseases. The intense inflammatory process in smokers culminates high levels of cytokines and oxidative stress and a great destruction of extracellular matrix and lung parenchyma cells, resulting in obstructive pulmonary disease [13]. In addition, the presence of oxidative damage in DNA, which can be 10

**Table 1. General characteristics of patients in the study.**

| Variables | Control (n = 18) | COPD without cancer (n = 18) | COPD with cancer (n = 18) | p Value |
|---|---|---|---|---|
| Gender (male/female) | 11/7 | 13/5 | 12/6 | 0.779 |
| Age (years) | $53.3 \pm 11.8^a$ | $65.0 \pm 10.7^b$ | $70.2 \pm 9.2^b$ | **<0.001** |
| GOLD I/II/III (n) | - | 5/8/5 | 4/7/7 | 0.572 |
| Current smokers (n/%) | 12/66 | 8/44 | 9/50 | 0.380 |
| Smoking history (pack-years) | $32.5 (17.0–60.0)^a$ | $49.0 (38.5–65.0)^{ab}$ | $69.0 (50.0–106.0)^b$ | **0.014** |
| CO (ppm) | 10.0 (0–18.0) | 1.5 (0.0–8.0) | 1.0 (0–3.0) | 0.071 |
| FVC (% predict) | $90.8 \pm 20.7$ | $84.1 \pm 20.9$ | $75.8 \pm 20.1$ | 0.101 |
| $FEV_1$ (% predict) | $88.3 \pm 21.2^a$ | $63.2 \pm 20.6^b$ | $59.6 \pm 20.2^b$ | **<0.001** |
| $FEV_1/FVC$ | $0.80 \pm 0.06^a$ | $0.57 \pm 0.08^b$ | $0.59 \pm 0.09^b$ | **<0.001** |
| BMI ($kg/m^2$) | $27.5 \pm 5.0^a$ | $22.0 \pm 4.7^b$ | $21.7 \pm 2.5^b$ | **<0.001** |
| FFM (kg) | $52.0 \pm 9.8$ | $44.2 \pm 11.4$ | $43.7 \pm 6.4$ | 0.062 |
| MRC | $0.5 \pm 0.6^a$ | $1.4 \pm 1.1^{ab}$ | $1.7 \pm 1.3^b$ | **0.011** |
| $SpO_2$ (%) | $97.0 (95.0–98.0)^a$ | $95.0 (93.0–97.0)^b$ | $94.0 (92.0–96.0)^b$ | **0.006** |

Values expressed as X ± SD or median (quartile 1—quartile 3). COPD I/II/III: mild/moderate/severe (GOLD: Global Initiative for Chronic Obstructive Lung Disease). FVC = forced vital capacity (% of predicted); $FEV_1$ = forced expiratory volume in the first second (% of predicted); BMI = body mass index; FFM = fat-free mass; ppm: parts per million; MRC = Modified Research Council; $SpO_2$ = pulse oximetry. a, b: different letters indicate statistically significant difference. $p < 0.05$. χ2, ANOVA or Kruskal-Wallis test.

times greater in smokers than in non-smokers, plays a role in the initiation of the tumorigenesis process and inactivation of defence mechanisms, such as low levels of antioxidants [13, 14].

MDA levels showed a positive association with the interaction between COPD and cancer, which indicates high levels of oxidative stress in these patients, capable of promoting DNA damage. MDA is a by-product of lipid peroxidation and is evaluated as a biomarker of oxidative stress in several pulmonary diseases, including COPD [30]. While there are few studies involving this biomarker, there is one study that suggests that patients with COPD presented higher levels of MDA in the exhaled breath condensate than the control group of smokers

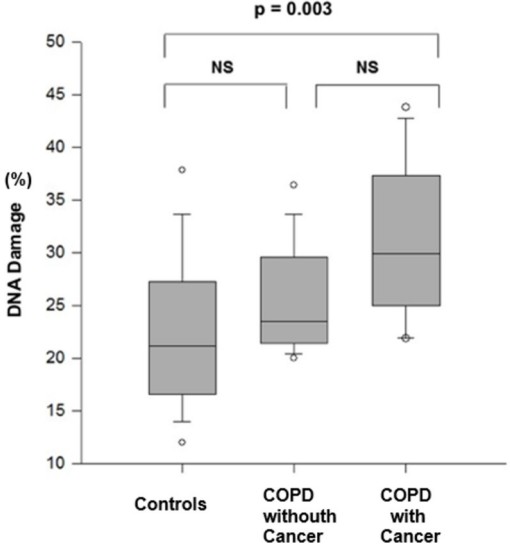

**Fig 2. Evaluation of DNA damage in controls and COPD patients with and without cancer (Kruskal-Wallis and Dunn's test).** NS: non-significant.

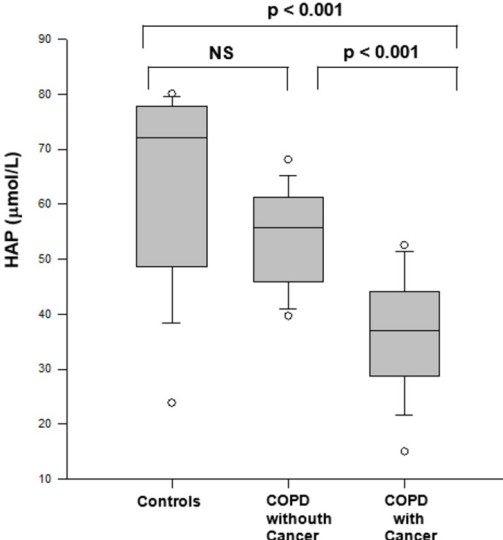

**Fig 3. Percentage of hydrophilic antioxidant capacity (HAP) in controls and COPD patients with and without cancer (Kruskal-Wallis and Dunn's test).** NS: non-significant.

[31]. In another study that analysed the serum MDA of patients with different lung diseases, the levels of this biomarker were significantly higher in the COPD group and lung cancer group than in the control group [15].

In the multiple linear regression analysis, DNA strand breaks also showed a positive association with the interaction between COPD and cancer. DNA damage involves breaks in the single and/or double strands in the genetic material and can be identified using the comet assay

**Table 2. Predictors of lipid peroxidation, antioxidant capacity, and DNA damage.**

| Dependent variable | Variables | Coefficient (95% IC) | p Value |
|---|---|---|---|
| MDA ($R^2 = 0.26$) | Age, years | 0.008 (-0.08; 0.83) | 0.969 |
| | Gender (male) | -0.18 (-2.83 0.45) | 0.150 |
| | Smoking (presence) | -0.25 (-2.83; 0.45) | 0.141 |
| | Smoking history (pack- years) | -0.07 (-0.02; 0.16) | 0.657 |
| | COPD (presence) | 0.24 (-0.54; 3.17) | 0.160 |
| | Interaction (COPDxCancer) | 0.45 (0.34; 4.69) | **0.025** |
| HAP ($R^2 = 0.57$) | Age, years | -0.96 (-0.54; 0.28) | 0.529 |
| | Gender (male) | 0.04 (-6.35; 9.43) | 0.696 |
| | Smoking (presence) | -0.23 (-16.11; 0.36) | 0.060 |
| | Smoking history (pack- years) | -0.17 (-0.18; 0.28) | 0.145 |
| | COPD (presence) | -0.29 (-19.25; -1.36) | **0.025** |
| | Interaction (COPDxCancer) | -0.70 (-36.9; -15.47) | **<0.001** |
| DNA strand breaks($R^2 = 0.40$) | Age, years | 0.21 (-0.09; 0.34) | 0.246 |
| | Gender (male) | -0.03 (-4.64; 3.68) | 0.818 |
| | Smoking (presence) | 0.29 (-0.12; 8.76) | 0.057 |
| | Smoking history (pack- years) | 0.13 (-0.30; 0.80) | 0.360 |
| | COPD (presence) | 0.18 (-1.80; 7.47) | 0.234 |
| | Interaction (COPDxCancer) | 0.50 (2.7–14.10) | **0.005** |

MDA: malondialdehyde; COPD: chronic obstructive pulmonary disease; HAP: Hydrophilic Antioxidant Performance; DNA: deoxyribonucleic acid.

[11, 32]. A recently published study compared DNA damage and the presence of cancerous markers between patients with lung cancer and COPD and patients with only lung cancer. Results showed that patients with COPD and cancer had an increase in the DNA damage marker levels compared to those with cancer but without COPD. However, in contrast to the findings of our research, active smoking was not a criterion for inclusion of samples, allowing statistically significant differences between groups in relation to tobacco use [33].

Our results showed that patients with COPD and lung cancer did not have a significant increase in DNA damage than that in the patients with COPD. Although these results indicate that the previous state of COPD without the initial presence of cancer does not determine further damage to DNA to influence the development of cancer, Tang et al. showed opposite results, inferring that the patients with lung cancer and COPD had increased levels of DNA damage markers than patients with lung cancer alone [33].

In the multiple linear regression analysis, the antioxidant capacity showed a negative association with the interaction between COPD and cancer and the presence of COPD, which shows that it further contributes in the pathophysiology of lung cancer in these patients. Patients with COPD had reduced level of antioxidants and increased level of oxidants in the lung as a result of cigarette use [34]. This oxidant/antioxidant imbalance is an important factor in lung pathogenesis as addressed by Rahman et al. in 2000. These authors showed that plasma antioxidant capacity was significantly decreased in smokers without lung disease and patients with COPD compared to non-smokers [35]. In a recent research conducted by Ahmad et al. in 2013, the results showed that the total antioxidant capacity in the plasma estimated as the ferric reducing ability was lower in patients with COPD than in the controls [36].

One limitation of this study is related to the average age of the control group, which was lower than that of the other groups. This fact may have culminated in differences in the levels of DNA damage and percentage of antioxidants from the natural process of aging, and not only in relation to the presence of COPD and the process of carcinogenesis [37]. In addition, the controls showed a higher BMI than patients with COPD with or without cancer. This may have been associated with high levels of DNA damage in multiple organs [38] in patients in the control group, approximating the results between them and the patients with COPD without lung cancer. In 2017, Gariballa et al. suggested that the measurement of waist circumference is a stronger predictor than the BMI; thus, our study findings may not have shown the effects of obesity accurately [39]. The patients with COPD (eutrophic) with lung cancer presented significantly higher levels of DNA damage than the controls (overweight), suggesting that the weight seemed to have a minimal influence on DNA damage. Additionally, we were not able to follow the clinical progression of the patients in the samples, which limited the possible association with the results found.

Finally, based on our results, patients with COPD and lung cancer presented increased levels of DNA damage compared to the controls. The MDA indicated high levels of oxidative stress in individual with COPD and lung cancer, but there was no difference compared to the others groups. The percentage of HAP was lower in patients who had an association of lung cancer and COPD compared to controls and patients with COPD without cancer. Further studies should be conducted to address the limitations of the present study. We recommend including selection criteria such as age, BMI, current smoker and, smoking history for better standardization for all individuals in the sample. We also recommend monitoring the clinical course of each individual in order to relate their current clinical stage to new levels of DNA damage and markers of oxidative stress.

## Author Contributions

**Conceptualization:** Mariana Gobbo Braz, Nayara Micarelli de Arruda, Laura Caram, Duelene Ludimila Nogueira, Suzana Erico Tanni, Irma de Godoy, Renata Ferrari.

**Data curation:** Mariana Gobbo Braz, Nayara Micarelli de Arruda, Laura Caram, Suzana Erico Tanni, Irma de Godoy, Renata Ferrari.

**Formal analysis:** Mariana Gobbo Braz, Nayara Micarelli de Arruda, Laura Caram, Duelene Ludimila Nogueira, Suzana Erico Tanni, Irma de Godoy, Renata Ferrari.

**Funding acquisition:** Suzana Erico Tanni, Irma de Godoy, Renata Ferrari.

**Investigation:** Mariana Gobbo Braz, Nayara Micarelli de Arruda, Laura Caram, Duelene Ludimila Nogueira, Suzana Erico Tanni, Irma de Godoy, Renata Ferrari.

**Methodology:** Mariana Gobbo Braz, Nayara Micarelli de Arruda, Laura Caram, Duelene Ludimila Nogueira, Suzana Erico Tanni, Irma de Godoy, Renata Ferrari.

**Project administration:** Laura Caram, Irma de Godoy, Renata Ferrari.

**Resources:** Mariana Gobbo Braz, Nayara Micarelli de Arruda, Laura Caram, Duelene Ludimila Nogueira, Suzana Erico Tanni, Irma de Godoy, Renata Ferrari.

**Supervision:** Mariana Gobbo Braz, Nayara Micarelli de Arruda, Laura Caram, Suzana Erico Tanni, Irma de Godoy, Renata Ferrari.

**Validation:** Mariana Gobbo Braz, Nayara Micarelli de Arruda, Laura Caram, Duelene Ludimila Nogueira, Suzana Erico Tanni, Irma de Godoy, Renata Ferrari.

**Visualization:** Camila Freitas dos Santos, Mariana Gobbo Braz, Nayara Micarelli de Arruda, Duelene Ludimila Nogueira, Suzana Erico Tanni, Irma de Godoy, Renata Ferrari.

**Writing – original draft:** Camila Freitas dos Santos, Mariana Gobbo Braz, Irma de Godoy, Renata Ferrari.

**Writing – review & editing:** Camila Freitas dos Santos, Mariana Gobbo Braz, Irma de Godoy, Renata Ferrari.

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
