## [Decision Letter · Decision Letter 0]

10 Aug 2022

PONE-D-22-00710

DNA damage and antioxidant capacity in COPD patients with and without lung cancer

PLOS ONE

Dear Dr. dos Santos,

Thank you for submitting your manuscript to PLOS ONE. After careful consideration, we feel that it has merit but does not fully meet PLOS ONE’s publication criteria as it currently stands. Therefore, we invite you to submit a revised version of the manuscript that addresses the points raised during the review process.

We look forward to receiving your revised manuscript.

Kind regards,

Emre Avci

Academic Editor

PLOS ONE

Journal Requirements:

Reviewers' comments:

Reviewer's Responses to Questions

**Comments to the Author**

1. Is the manuscript technically sound, and do the data support the conclusions?

Reviewer #1: Yes

Reviewer #2: Partly

Reviewer #3: Yes

2. Has the statistical analysis been performed appropriately and rigorously? 

Reviewer #1: Yes

Reviewer #2: Yes

Reviewer #3: Yes

3. Have the authors made all data underlying the findings in their manuscript fully available?

Reviewer #1: Yes

Reviewer #2: Yes

Reviewer #3: Yes

4. Is the manuscript presented in an intelligible fashion and written in standard English?

Reviewer #1: Yes

Reviewer #2: Yes

Reviewer #3: Yes

5. Review Comments to the Author

Reviewer #1: The manuscript by Freitas dos Santos et al. deals with COPD patients and their vulnerability to lung cancer due to increased DNA damage and altered antioxidant status. Obtained results may provide a contribution to cancer studies. The aim of the study is remarkable. Although findings are not sufficient to claim exact correlation between cancer and oxidative status of COPD patients, in my opinion, the scientific level of the manuscript is appropriate for PlosOne in its current form.

Reviewer #2: It is a valuable study which is investigating the DNA damage and oxidative stress and their relationship in patients with COPD with and without lung cancer. But I have some suggestions that I have stated below.

Introduction:

The importance of oxidative stress markers in cancer should be briefly mentioned.

Material and Methods:

The selection criteria of the control group are not clear from the text, more detailed information should be given.

Why MDA and HAP have been as markers? Could you explain with references?

Results:

In Table 1, too many spirometry results have been shared, although they are not mentioned in the text. Please simplify the data in table 1

Are there associations between the number and duration of smoking and the results? It should be analyzed

Is there an association between the clinical progression and the results? It should be analyzed, if not possible please specified as a limitation.

Discussion:

Analyzes have been performed according to the information about age, gender etc. and shared in the tables by authors. But have not been mentioned in the discussion part. Please, mention briefly.

The discussion has been based on smoking by the authors. But, correlation analysis with smoking has not been performed. İmprove the discussion section according to analysis suggested in the results section.

What are the authors' recommendations regarding the results obtained with limited patient data? Please improve the conclusion part at the end of the manuscript.

Reviewer #3: The manuscript is well written. Authors have evaluated the DNA damage in COPD and COPD with cancer patients in comparison to control. The results were presented with appropriate statistical evaluation.

6. PLOS authors have the option to publish the peer review history of their article (what does this mean?). If published, this will include your full peer review and any attached files.

Reviewer #1: No

Reviewer #2: No

Reviewer #3: No

---

## [Author Response · Author response to Decision Letter 0]

31 Aug 2022

We appreciate reviewer’ comments concerning our manuscript entitled: “DNA damage and antioxidant capacity in COPD patients with and without lung cancer”. Those comments are all valuable and very helpful for revising and improving our paper. Based on your comment and request, we have made modification in the original manuscript. We attached the revised manuscript in a separate file entitled 'Revised Manuscript with Track Changes' and we uploaded a separate file entitled 'Manuscript' without tracked changes. We hope that our manuscript is now acceptable for publication in this renowned leading journal.

---

## [Editor Report · Decision Letter 1]

26 Sep 2022

DNA damage and antioxidant capacity in COPD patients with and without lung cancer

PONE-D-22-00710R1

Dear Dr. dos Santos,

We’re pleased to inform you that your manuscript has been judged scientifically suitable for publication and will be formally accepted for publication once it meets all outstanding technical requirements.

Kind regards,

Emre Avci

Academic Editor

PLOS ONE

---

## [Editor Report · Acceptance letter]

25 Oct 2022

PONE-D-22-00710R1 

DNA damage and antioxidant capacity in COPD patients with and without lung cancer 

Dear Dr. dos Santos:

I'm pleased to inform you that your manuscript has been deemed suitable for publication in PLOS ONE. Congratulations! Your manuscript is now with our production department. 

Kind regards, 

on behalf of

Dr. Emre Avci 

Academic Editor

PLOS ONE